# Preoperative Motor Function Associated with Short-Term Gain of Health-Related Quality of Life after Surgery for Lumbar Degenerative Disease: A Pilot Prospective Cohort Study in Japan

**DOI:** 10.3390/healthcare11243103

**Published:** 2023-12-05

**Authors:** Yuya Ishibashi, Yosuke Tomita, Shigeyuki Imura, Nobuyuki Takeuchi

**Affiliations:** 1Department of Physical Therapy, Graduate School of Health Care, Takasaki University of Health and Welfare, Takasaki 370-0033, Gunma, Japan; 2030201@takasaki-u.ac.jp (Y.I.); imura-s@jcom.home.ne.jp (S.I.); ntakeuchi@takasaki-u.ac.jp (N.T.); 2Department of Rehabilitation, Harunaso Hospital, Takasaki 370-3347, Gunma, Japan; 3Department of Medical Device Development, mediVR, Inc., Chuo-ku 103-0022, Tokyo, Japan

**Keywords:** lumbar surgery, quality of life, minimal clinically important difference, motor function

## Abstract

This study aimed to estimate the relationship between preoperative motor function and short-term recovery of health-related quality of life after lumbar surgery in patients with lumbar degenerative disease. This prospective cohort study involved 50 patients with lumbar degenerative disease at a general hospital in Japan. The primary outcome was the achievement of minimal clinically important difference (MCID) for EuroQOL 5 dimensions (EQ-5D) at discharge. Preoperative demographic, medication, surgical, and physical function data were collected. Logistic regression analysis was performed using the achievement of MCID for EQ-5D as the dependent variable and preoperative characteristics, including the Five Times Sit to Stand test (FTSTS), Oswestry Disability Index (ODI), and Self-rating Depression Scale (SDS), as the independent variables. The logistic regression analysis showed that Model 1 had a moderate predictive accuracy (Nagelkerke R^2^: 0.20; Hosmer–Lemeshow test: *p* = 0.19; predictive accuracy: 70.0%). Among the independent variables in the logistic regression model, the FTSTS was the only independent variable related to the achievement of MCID for EQ-5D at discharge (odds ratio: 0.03; 95% CI: 1.79 × 10^−3^, 0.18). Our results highlighted the importance of baseline motor function in the postoperative recovery of health-related quality of life in individuals with lumbar degenerative disease.

## 1. Introduction

Lumbar degenerative disease, also known as Degenerative Lumbar Spine Disease, encompasses conditions such as Lumbar Disk Herniation (LDH), Lumbar Spinal Stenosis (LSS), and Degenerative Spondylolisthesis (DS). Globally, over 266 million individuals have been diagnosed with lumbar degenerative disease [1]. This condition not only manifests as lower back pain but can also give rise to motor impairments [2], gait instability [3], and diminished physical activity [4]. Consequently, these symptoms contribute to a reduction in activities of daily living (ADL) and a decline in health-related quality of life (HRQOL).

Lumbar surgery is one of the most effective treatments to improve HRQOL, a primary goal of medical interventions in patients with lumbar degenerative disease. Studies have shown that the extent of improvement achieved by lumbar surgery depends on preoperative characteristics and can be predicted using baseline patient characteristics. For example, the extent of HRQOL at 2 years post lumbar surgery, evaluated by the EuroQOL 5 dimensions (EQ-5D) score, can be predicted by the baseline score of EQ-5D [5]. Similarly, the EQ-5D score at 6 and 24 months post lumbar surgery can be predicted by the baseline scores of EQ-5D, activities of daily living (ADL; Oswestry Disability Index [ODI]), and mental status (mental component score of the Short-Form 36-Item Health Survey [SF-36]) [6]. The achievement of minimal clinically important difference (MCID) for EQ-5D at 1-year post-surgery is related to the diagnosis of depression and the preoperative score of EQ-5D [7]. Abbott et al. (2010) also showed that the relationship between pain intensity, HRQOL, and ADL was mediated by emotional and cognitive factors such as pain perception, coping strategy, and catastrophizing [8]. However, none of the predictive models introduced above included independent variables related to motor function, which is often reduced in patients who undergo lumbar surgery.

Recovery of HRQOL and ADL after surgery may be related to motor function in patients with lumbar degenerative disease. For example, the improvement of the self-reported physical function (physical component score of the 12-item Short Form Survey) is related only to the motor-related doman of the ODI, such as “Lifting”, “Standing”, and “Traveling” [9]. Another study reported that predictive accuracy of ADL function increased when variables related to motor function were included as independent variables in addition to patient characteristics and pain intensity [10]. Considering that reduced motor function is one of the primary symptoms in patients with a lumbar degenerative disease [11,12], a detailed evaluation of motor function may help predict postoperative recovery of HRQOL. However, the majority of prior investigations into preoperative and postoperative patient characteristics have predominantly relied on patient-reported outcomes (PROs). Standard PROs, such as EQ-5D and ODI, have been extensively used in various registries, such as the Swedish Spine Registry [13] and the Norwegian Registry for Spine Surgery [14]. Although these PROs enable clinicians to evaluate the client’s perspective, they are susceptible to response bias [15] and may not accurately reflect the client’s motor function [2]. To comprehensively capture the patient’s condition, it is important to complement PROs with clinical assessments based on objective evaluations by clinicians. Utilizing measures such as the Five Times Sit to Stand Test (FTSTS) and the Timed “Up and Go” test (TUG) can provide a more holistic understanding of the patient’s condition, addressing the limitations associated with PROs [16]. Lumbar degenerative disease results in diverse motor impairments, including paraspinal muscle atrophy [17], diminished static [18] and dynamic balance function [19,20], and reduced walking endurance [21]. Previous studies have employed clinical evaluations such as the TUG [19], FTSTS [20], and 6-min walk distance (6MWD) [21] to objectively assess motor impairments in individuals with lumbar degenerative disease. These assessment tools are not mutually exclusive, as they collectively reflect the client’s mobility. However, each assessment may elucidate different aspects of subsystems contributing to overall mobility. Despite these prior investigations, the relationship between preoperative assessment outcomes and postoperative recovery of HRQOL in individuals with lumber degenerative disease remains unclear [19,20,21].

Therefore, the objective of this study was to estimate the relationship between preoperative motor function and short-term recovery of HRQOL after lumbar surgery in patients with lumbar degenerative disease. We hypothesized that adding independent variables related to preoperative motor function to an existing predictive model would improve the model’s predictive accuracy in estimating the achievement of MCID after lumbar surgery.

## 2. Materials and Methods

### 2.1. Study Design and Patients

Patient selection flow is shown in Figure 1. In this prospective cohort study, we initially included 108 patients who underwent surgery for lumbar degenerative disease at a general hospital in Japan between May 2021 and October 2021. Patients were excluded when they could not complete clinical evaluations due to cognitive disorders, respiratory problems, and/or severe pain (n = 3). Patients were also excluded if they had lumbar surgery for idiopathic scoliosis, spinal tumor, and spinal cord injury (n = 20); had a history of lumbar surgery (n = 16); did not provide consent for study participation (n = 7); had administrative difficulties due to postoperative schedule (n = 8); were aged under 20 (n = 1); and had a distinctively different postoperative clinical course (n = 1). Follow-up data were not collected for two patients. Finally, 50 consecutive participants were enrolled in the study. After explaining the study procedures and risks involved in the study, written consent was obtained from all participants. There was a potential risk of injury or falls due to excessive effort during the clinical assessments. To minimize the risk, the examiner provided detailed explanation regarding the clinical assessment procedures. Furthermore, vigilant monitoring of the patient was consistently maintained to promptly identify any indications of excessive effort or potential hazards associated with the assessments. The study was approved by the research ethics committee of Takasaki University of Health and Welfare (approval number: 2065). This study followed the Strengthening the Reporting of Observational Studies in Epidemiology criteria.

### 2.2. Primary Outcome

The primary outcome was the achievement of minimal clinically important difference (MCID) of EQ-5D score (achieved vs. not achieved) at discharge. The EQ-5D is a self-administered questionnaire that measures five dimensions of health, including mobility, self-care, usual activities, pain and discomfort, and anxiety and depression [22]. EQ-5D can reflect the health status of patients with chronic low back pain and degenerative disc disease more broadly compared to the Short Form 6 Dimension [23]. We used the three-level version of the EQ-5D, and the score was converted to a utility value. The ΔEQ-5D was calculated by subtracting the preoperative EQ-5D utility value from the value at discharge. The MCID of the EQ-5D in this study was set as 0.20 based on a previous study [24].

### 2.3. Exposures

Demographic, medication, surgical, and clinical data were collected at 1–3 days before the surgery. The measured variables included age; sex; height; body mass; body mass index (BMI) [25]; history of smoking [26,27]; duration of postoperative hospital stay; history of falling in the past 6 months; diagnosis of degenerative spondylosis, degenerative spondylolisthesis, lumbar spinal stenosis, and adult spinal deformity; diagnosis of diabetes mellitus, asthma, and osteoarthritis; preoperative number of medications; surgery type (discectomy, fenestration, instrumented fusion, lumbosacral fusion); number of fused segments; number of decompressed segments; analgesic use [28]; and adjuvant analgesic use. Clinical data included intensity of back and leg pain; duration of symptoms [29]; Oswestry Disability Index (ODI) [30]; Self-rating Depression Scale (SDS) [31,32]; 10 Meter Walking Test; Timed “Up and Go” Test (TUG) at comfortable gait speed [19,33]; 6MWD [21]; Knee Extension Strength with Handheld Dynamometer; and FTSTS [20]. Preoperative ODI score was used as a predictor of EQ-5D score postoperative 12 months in patients with lumbar degenerative disease [34]. Similarly, preoperative SDS scores predicted achievement of MCID for SF-36 at 12 months postoperatively [32]. We used TUG [19] and FTSTS [20] as the objective measures of motor function. These assessments provided insight into objective functional impairments by encompassing motor tasks commonly challenging for individuals with lumbar degenerative disease, such as gait and sit-to-stand transitions [19,20].

### 2.4. Statical Analysis

In univariate analysis, we compared demographics, medication, surgical, and physical function data between two groups (MCID achieved vs not achieved). We used the independent *t*-test, Mann–Whitney test, and Chi-square test for the comparisons of continuous variables with normal distribution, those without normal distribution, and categorical variables, respectively. We also computed the area under the receiver operating characteristics curves (AUROC) and its 95% confidence interval (CI) for continuous variables with significant differences. The cut-off value was identified using the Youden Index [35].

In multivariable analysis, a multivariable logistic regression model was computed using the achievement of MCID at discharge as the dependent variable and preoperative clinical scores as the independent variables, including SDS, FTSTS, and ODI (Model 1). The independent variables of the Model 1 were selected according to the clinical relevance judged by researchers based on the results of previous studies (Table 1). We added FTSTS to the existing models to infer the impact of pre-operative motor function to the outcome. The goodness-of-fit of the created logistic model was evaluated using the likelihood ratio test, Nagelkerke R^2^, overall accuracy, and the Hosmer–Lemeshow test. We also created a multiple regression model (Model 2), with ΔEQ-5D at discharge as the dependent variable and the same independent variables as Model 1. The goodness-of-fit of the multiple regression model was evaluated using the F value and adjusted R^2^ value. The absence of multicollinearity was verified using the correlation coefficient and variance inflation factor (VIF). The distribution of residual error in the multiple regression analysis was evaluated using the Durbin–Watson and Shapiro–Wilk tests. Statistical analyses were performed using SPSS v. 23.0 (IBM Corp., Armonk, NY, USA). A two-tailed *p*-value of <0.05 was considered statistically significant.

## 3. Results

Patient recruitment flow is shown in Figure 1. We initially included 108 subjects admitted for lumber surgery during the study period. We excluded 56 patients based on our exclusion criteria. Two patients declined to participate in the study, and another did not complete the follow-up evaluation. Finally, we included 50 patients in the analysis. The demographic characteristics of the 50 patients are presented in Table 2 and Table 3. Data are presented separately for MCID achieved (n = 22/50, 44.0%) and not achieved (n = 28/50; 56.0%) at discharge. The univariate analysis showed that FTSTS was significantly lower in the MCID achieved group than in the not achieved group (Table 3). The ROC analysis showed that the FTSTS of 11.4 s had the highest discriminative accuracy (sensitivity: 40%; specificity: 96.2%).

Multivariable regression analysis showed that Model 1 (multivariable logistic regression model) had a moderate predictive accuracy (Table 4; Nagelkerke R^2^: 0.20; Hosmer–Lemeshow test: *p* = 0.19; predictive accuracy: 70.0%). Among the independent variables in Model 1, the FTSTS was the only independent variable significantly related to the achievement of MCID for EQ-5D at discharge (odds ratio: 0.03; 95% CI: 1.79 × 10^−3^, 0.18). Model 2 (Table 5; multivariable linear regression model) also had a moderate predictive accuracy (F = 3.15; adjusted R^2^ = 0.12; *p* = 0.03). Among the independent variables in Model 2, FTSTS was the only independent variable significantly related to a change of EQ-5D at discharge (B: −0.13; 95% CI: −0.23, −0.35). In Model 2, low VIF indicated the absence of multicollinearity between independent variables.

## 4. Discussion

The objective of this study was to estimate the relationship between preoperative motor function and short-term recovery of HRQOL after lumbar surgery in patients with lumbar degenerative disease. Our results showed that adding preoperative FTSTS to the model with depressive state and ADL function as the independent variables allowed us to predict changes in HRQOL with moderate precision, where FTSTS was the only predictor significantly related to the postoperative recovery of EQ-5D. Our results highlighted the importance of a baseline motor function to achieve meaningful change in HRQOL in individuals with lumbar degenerative disease.

Our results demonstrated that the postoperative change of EQ-5D was greater in patients with lower preoperative motor function. Existing statistical models to predict postoperative EQ-5D score and its change from the preoperative score are summarized in Table 1. There are no statistical models that include motor function as the independent variable. A recent systematic review showed that the improvement of EQ-5D and ODI after surgery was greater in patients with lower baseline scores of these measures in patients who underwent lumbar disc discectomy [36]. At the same time, the relationship of the preoperative motor function with the recovery of these scores remained unknown. One cross-sectional observational study reported that the FTSTS was moderately correlated with HRQOL in patients with lumbar degenerative diseases, consistent with our study results [20]. FTSTS may be a sensitive functional measurement to reflect functional impairments in patients with lumbar degenerative diseases since the repetition of sit-to-stand movement requires lower limb muscle strength, somatosensory function, balance adjustments, and psychological factors. However, a recent randomized controlled trial showed that preoperative physiotherapy sessions improved preoperative motor function and HRQOL compared to the control group, while postoperative change of HRQOL was not different between the intervention and control groups [37]. Therefore, although our study results showed that preoperative motor function is related to the postoperative recovery of HRQOL even when adjusted for preoperative ADL function and depressive state, the causal relationship between these factors may not be strong. In future studies, it is important to understand factors that may mediate the relationship between preoperative motor function and postoperative recovery of HRQOL.

Our study participants had relatively higher motor preoperative HRQOL and motor function than those in previous studies. For example, Nayak et al. (2019) reported that participants in the study had a mean EQ-5D score of 0.370~0.420 [38], while participants in our study had higher scores (MCID achieved: 0.530 and not achieved: 0.590). Similarly, the mean FTSTS in patients with lumbar degenerative diseases reported in a previous study was 13.3 ± 7.9 s, which was also worse than the performance of our study participants [20]. These results suggest that our study participants had higher EQ-5D and FTSTS scores than in previous studies. Therefore, preoperative factors related to the postoperative recovery of EQ-5D may differ in patients with more severe motor impairments. However, our study results shed light on the importance of preoperative motor performance in predicting the recovery of HRQOL, even in patients with relatively good motor performance. Further study with a larger sample size is needed to perform subgroup analyses according to different baseline characteristics to understand predictors for the postoperative recovery of HRQOL in this population.

Our study had several limitations. First, the sample size was limited to 50 participants. The small sample size only allowed us to use three independent variables based on previous studies. Further study is needed to examine the relationship of preoperative motor performance with postoperative HRQOL. Second, our study participants’ HRQOL and motor performance were relatively high compared to previous studies. The study results cannot be generalized to patients with more severe motor performance. Third, the external validity of the created regression models was not examined in this study. External validation is needed to evaluate the robustness of the proposed models.

## 5. Conclusions

This prospective cohort study found that preoperative motor function was significantly related to the postoperative recovery of EQ-5D in patients with lumbar degenerative disease, even when adjusted for preoperative ADL function and depressive status. Patients with lower preoperative motor function (FTSTS > 11.4 s) were likely to achieve the MCID for EQ-5D at discharge after the lumbar surgery. Our results highlighted the importance of baseline motor function in the postoperative recovery of HRQOL in individuals with lumbar degenerative disease. Further study with a greater sample size is needed to clarify the relationship between preoperative motor function and postoperative HRQOL.

## Figures and Tables

**Figure 1 healthcare-11-03103-f001:**
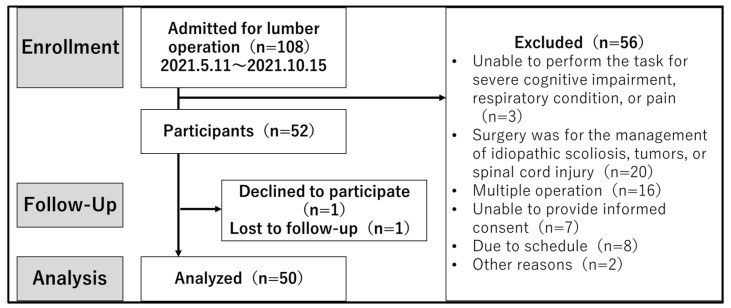
Flowchart depicting the patient selection process.

**Table 1 healthcare-11-03103-t001:** Regression models to predict postoperative EQ-5D score and its change from the preoperative score.

Study	Outcome Measure	Observation Period	Independent Variables
Abbott et al. 2010 [8] n = 107	EQ-5D	2 year–3 year	Baseline of The Coping Strategy; questionnaire’s control over pain subscale; baseline of Back Beliefs Questionnaire.
Silverplats et al. 2011 [5] n = 117	EQ-5D	2 year	Baseline of EQ-5D.
Hey et al. 2018 [6] n = 292	ΔEQ-5D	6 m, 1 year	Baseline of EQ-5D; baseline of Mental Component Score from Short-Form 36-Item Health Survey.
McGirt et al. 2017 [34] n = 7618	EQ-5D	1 year	Age; female; BMI; ASA grade; education status; occupation status; race status; insurance status; current smoker; baseline of ambulation status (used an assist device); baseline of EQ-5D; baseline of ODI; surgical approach; baseline of leg pain (NRS); baseline of back pain (NRS); symptom duration; predominant.pain.location; psychological distress (none/moderately/severely).
Devin et al. 2020 [29] n = 22,022	ΔEQ-5D	1 year	Age; BMI; education status; ASA grade; insurance status; race status; worker’s compensation; current smoker; surgical approach (lateral only); history of anxiety; baseline of leg pain (NRS); symptom duration; baseline of back pain (NRS); predominant pain location; baseline of ambulation status (used an assist device).
Alentado et al. 2017 [7] n = 231	Achievement of MCID for EQ-5D	1 year	Baseline of EQ-5D; surgery approach (L5-S1 fusion); diagnosis of depression; diagnosis of lumbar spondylolisthesis.

EQ-5D: EuroQOL-5 Dimension three-level version utility value; BMI: Body Mass Index; ASA: American Society of Anesthesiologists; MCID: Minimal clinically important difference; NRS: Numerical rating scale.

**Table 2 healthcare-11-03103-t002:** Demographic characteristics of the participants (n = 50).

Variables	Achievement of MCID for EQ-5D at Discharge			*p*-Value
	Achieved (n = 22, 44%)	Not Achieved (n = 28, 56%)	95%CI	
Age (y), mean (SD)	66.5 (13.8)	65.2 (11.6)	−5.95, 8.52	0.361
Gender, n (%)				
Female	9 (40.9)	8 (28.6)	0.53, 5.64	0.361
BMI (kg/m^2^), mean (SD)	24.0 (3.9)	25.6 (3.6)	−3.79, 0.45	0.722
History of smoking, n (%)	9 (42.9)	17 (60.7)	0.15, 1.53	0.215
Hospital stays (day), mean (SD)	26.2 (21.7)	22.8 (18.1)	−7.86, 14.81	0.465
History of Falls, n (%)	3 (13.6)	6 (21.4)	0.13, 2.63	0.371
Diagnosis				
Disc herniation, n (%)	4 (18.2)	4 (14.3)	0.96, 1.15	0.502
Degenerative spondylosis, n (%)	1 (4.5)	4 (14.3)	0.03, 2.76	0.259
Degenerative spondylolisthesis, n (%)	8 (36.4)	8 (28.6)	0.43, 4.72	0.558
Spinal stenosis, n (%)	18 (81.8)	25 (89.3)	0.11, 2.72	0.362
Adult spinal deformity, n (%)	3 (13.6)	2 (7.1)	0.31, 13.51	0.384
Comorbidity				
Osteoarthritis, n (%)	2 (9.1)	5 (17.9)	0.23, 31.90	0.322
Asthma, n (%)	1 (4.5)	1 (3.6)	0.08, 21.78	0.691
Diabetes mellitus, n (%)	3 (13.6)	7 (25.0)	0.11, 2.10	0.263
Medication				
Number of medications, mean (SD)	4.4 (3.7)	5.0 (4.0)	−2.78,1.67	0.716
Analgesic use, n (%)	11 (50.0)	14 (50.0)	0.33, 3.06	1.000
Adjuvant analgesic use, n (%)	7 (31.8)	7 (25.0)	0.41, 4.84	0.594
Surgery Status				
No. of fused segments, median (IQR)	1 (2)	1 (1.8)	−1.00, 1.00	0.894
No. of decompressed segments, median (IQR)	2 (2)	1 (1)	−1.00, 0.00	0.117
Discectomy, n (%)	2 (9.1)	3 (10.7)	0.13, 5.48	0.616
Fenestration, n (%)	5 (22.7)	6 (21.4)	0.28, 4.14	0.589
Instrumented fusion, n (%)	15 (68.2)	19 (67.9)	0.31, 3.36	0.981
Lumbosacral fusion, n (%)	5 (22.7)	7 (25.0)	0.24, 3.28	0.852

MCID: Minimal clinically important difference; 95%CI: 95% confidence interval; EQ-5D: EuroQOL-5 Dimension three-level version utility value; BMI: Body mass index; SD: Standard Deviation; IQR: Interquartile Range.

**Table 3 healthcare-11-03103-t003:** Clinical outcome and motor function of the participants before the surgery (n = 50).

Variables	Achievement of MCID for EQ-5D at Discharge			*p*-Value
	Achieved (n = 22, 44%)	Not Achieved (n = 28, 56%)	95%CI	
Patient-Reported Outcome				
EQ-5D, median (IQR)	0.53 (0.23)	0.59 (0.11)	0.00, 0.163	0.060
EQ-VAS, mean (SD)	53.2 (19.9)	58.3 (15.7)	−15.28, 4.97	0.311
ODI (0–100), mean (SD)	44.9 (19.0)	39.8 (15.9)	−4.74, 15.12	0.298
Severity of symptoms				
Back Pain (NRS), median (IQR)	2.5 (5.0)	3.5 (7.8)	0.00, 3.00	0.242
Leg Pain (NRS), median (IQR)	6.5 (6.0)	6.5 (8.3)	−3.00, 2.00	0.759
Leg Numbness (NRS), median (IQR)	4.5 (4.3)	5.0 (6.8)	−1.00, 3.00	0.484
Duration of Symptoms (Month), median (IQR)	91.2 (132.2)	98.5 (131.1)	−24.00, 32.00	0.845
Psychological status				
SDS (20–80), mean (SD)	41.9 (10.4)	40.8 (9.8)	−4.66, 6.89	0.700
<48 points, n (%)	15 (68.2)	22 (78.6)	0.48, 6.11	0.406
Motor function				
Fast gait speed (m/s), mean (SD)	1.67 (0.46)	1.77 (0.46)	−0.37, 0.19	0.508
Step length (meter), mean (SD)	0.68 (0.13)	0.71 (0.13)	−0.11, 0.05	0.390
TUG at comfortable speed (second), mean (SD)	12.0 (6.9)	9.0 (2.1)	−0.21, 6.11	0.066
Walking distance (m), mean (SD)	377.5 (154.0)	387.6 (132.6)	−98.37, 76.99	0.818
Knee Extension Strength (kgf), mean (SD)	27.9 (10.1)	30.5 (10.8)	−8.67, 3.57	0.406
FTSTS (second), mean (SD)	11.0 (5.8)	7.8 (2.2)	0.36, 6.01	0.029
Foot Tapping Test (times), mean (SD)	26.1 (8.0)	29.1 (8.1)	−7.58, 1.71	0.141

MCID: Minimal clinically important difference; EQ-5D: EuroQOL-5 Dimension three-level version utility value; 95%CI: 95% confidence interval; SD: Standard Deviation; IQR: Interquartile Range; NRS: Numeric Rating Scale; EQ-VAS: EuroQOL-Visual Analog Scale; ODI: Oswestry disability index; SDS: Self-rating Depression Scale; TUG: Timed “Up and Go” Test; FTSTS: Five Times Sit to Stand Test; *p*-value of <0.05 is highlighted in bold.

**Table 4 healthcare-11-03103-t004:** Results of multivariate logistic regression models to predict achievement of MCID for EQ-5D at discharge.

Model No.	Independent Variables	Odds Ratio (95%CI)	*p*-Value	AUROC (95%CI)	Overall Accuracy (%)	Sensitivity (%)	Specificity (%)
Model 1	FTSTS ≤ 11.41 s	0.03 (1.79 × 10^−3^, 0.48)	0.013	0.674 (0.52, 0.83)	70.0	89.2	45.4
(n = 50)	ODI	1.01 (0.97, 1.05)	0.754				
	SDS < 48 points	1.37 (0.25, 7.62)	0.716				

MCID: Minimal clinically important difference; EQ-5D: EuroQOL-5 Dimension three-level version utility value; 95%CI: 95% confidence interval; AUROC: Area under the receiver operating characteristic curve; FTSTS: Five Times Sit to Stand Test; ODI: Oswestry Disability Index; SDS: Self-rating depression scale; *p*-value of <0.05 is highlighted in bold.

**Table 5 healthcare-11-03103-t005:** Results of multivariate regression models to change of EQ-5D at discharge.

Model No.	Independent Variables	B (95%CI)	β	*p*-Value	VIF
Model 2	FTSTS ≤ 11.41 s	−0.13 (−0.23, −0.35)	−0.40	0.009	1.20
(n = 50)	ODI	1.12 × 10^−3^ (−1.59 × 10^−3^, 3.82 × 10^−3^)	0.13	0.410	1.36
	SDS < 48 points	0.05 (−0.60, 0.16)	0.14	0.375	1.42

EQ-5D: EuroQOL-5 Dimension three-level version utility value; B: Unstandardized partial regression coefficient; 95%CI: 95% confidence interval; β: standardized partial regression coefficient; VIF: Variance inflation factor; FTSTS: Five Times Sit to Stand Test; ODI: Oswestry Disability Index; SDS: Self-rating depression scale; *p*-value of <0.05 is highlighted in bold.

## Data Availability

The data presented in this study are available on request from the corresponding author.

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
