# Peer review of "Preoperative Motor Function Associated with Short-Term Gain of Health-Related Quality of Life after Surgery for Lumbar Degenerative Disease: A Pilot Prospective Cohort Study in Japan"

_healthcare, 2023, doi:10.3390/healthcare11243103_

Round 1
Reviewer 1 Report
Comments and Suggestions for Authors
Dear Authors
From my review, I make the following observations:
Title: Would be helpful to include the location where the study was conducted.
Introduction: I would say this could work. Perhaps they could say the reason to consider those instruments for measuring outcome (EQ-5D, ODI, SDS, TUG, ETC) over others. Is it possible to have references stating that there are similarities with others?
Materials and methods: the design of the study has to be improved. In line 73 talk about risk "involved in the study", explicit the risk.
As you say in the limitation, the sample size is really small.
Results: In all the Tables add the IC and use bold to sign the significant results; add the adjustments that have been made.
Discussions: why is table 5 in this section? could be better in results. However, some studies are dated.
References: the bibliography is dated and should be updated. It would be useful to include the references of these articles for international comparisons:
- Knott RJ, Black N, Hollingsworth B, Lorgelly PK. Response-Scale Heterogeneity in the EQ-5D. Health Econ. 2017 Mar;26(3):387-394. doi: 10.1002/hec.3313. Epub 2016 Jan 12. PMID: 26756822.
- Fineschi D, Acciai S, Napolitani M, Scarafuggi G, Messina G, Guarducci G, Nante N. Game of Mirrors: Health Profiles in Patient and Physician Perceptions. International Journal of Environmental Research and Public Health. 2022; 19(3):1201. https://doi.org/10.3390/ijerph19031201
- Johnsen LG, Hellum C, Nygaard OP, Storheim K, Brox JI, Rossvoll I, Leivseth G, Grotle M. Comparison of the SF6D, the EQ5D, and the oswestry disability index in patients with chronic low back pain and degenerative disc disease. BMC Musculoskelet Disord. 2013 Apr 26;14:148. doi: 10.1186/1471-2474-14-148. PMID: 23622053; PMCID: PMC3648434.
Comments on the Quality of English LanguageMinor editing of English language required
Author Response
We would like to express our gratitude to the reviewer for conducting a thorough review and for providing constructive comments and suggestions. In response, we have addressed the comments and highlighted the changes in red within the revised manuscript. Please find our detailed point-by-point response to each comment below.
Comment 1
Title: Would be helpful to include the location where the study was conducted.
>> Response
In accordance with the reviewer's recommendation, we have explicitly indicated that the study was conducted in Japan. This information is now clearly presented in the title, abstract (page 1, line 17), and the materials and methods section (page 3, line 97).
Comment 2
Introduction: I would say this could work. Perhaps they could say the reason to consider those instruments for measuring outcome (EQ-5D, ODI, SDS, TUG, ETC) over others. Is it possible to have references stating that there are similarities with others? 
>> Response
As suggested by the reviewer, we have incorporated the rationale for the outcome measures and their measurement properties into both the Introduction and Materials & Methods sections, as follows:
1.Introduction
“However, the majority of prior investigations into preoperative and postoperative pa-tient characteristics have predominantly relied on patient-reported outcomes (PROs). Standard PROs such as EQ-5D and ODI have been extensively used in various registries, such as the Swedish Spine Registry [13] and the Norwegian Reg-istry for Spine Surgery [14]. Although these PROs enable clinicians to evaluate client’s perspective, they are susceptible to response bias [15] and may not accurately reflect the client’s motor function [2]. To comprehensively capture the patient’s condition, it is important to complement PROs with clinical assessments based on objective evaluations by clinicians. Utilizing measures such as the Five Times Sit to Stand Test (FTSTS) and the Timed “Up and Go” test (TUG) can provide a more holistic understanding of the patient’s condition, addressing the limitations associated with PROs [16]. Lumbar degenerative disease results in diverse motor impairments, including paraspinal muscle atrophy [17], diminished static [18] and dynamic balance function [19, 20], and reduced walking endurance [21]. Previous studies have employed clinical evaluations such as the TUG [19], FTSTS [20], and 6-minute walk distance (6MWD) [21] to objectively assess motor impairments in in-dividuals with lumbar degenerative disease. These assessment tools are not mutually exclusive, as they collectively reflect the client’s mobility. However, each assessment may elucidate different aspects of subsystems contributing to overall mobility. Despite these prior investigations, the relationship between preoperative assessment outcomes and postoperative recovery of HRQOL in individuals with lumber degenerative disease remains unclear [19, 20, 21].” (page 2, line 65-85)
2.2. Primary Outcome
“EQ-5D can reflect the health status of patients with chronic low back pain and degen-erative disc disease more broadly compared to the Short Form 6 Dimension [23].” (page 3, line 118-120)
2.3. Exposures
“Preoperative ODI score is used as a predictor of EQ-5D score postoperative 12 months in patients with lumbar degenerative disease [34]. Similarly, preopera-tive SDS scores predict achievement of MCID for SF-36 at 12 months postoperatively [32]. We used TUG [19] and FTSTS [20] as the objective measures of motor function. These assessments provide insight into objective functional impairments by encompassing motor tasks commonly challenging for indi-viduals with lumbar degenerative disease, such as sit-to-stand transitions and gait [19, 20].” (page 3, line 137-143)
Comment 3
Materials and methods: the design of the study has to be improved. In line 73 talk about risk "involved in the study", explicit the risk.
>> Response
We appreciate the reviewer for highlighting an important point. Acknowledging the lack of clarity in describing the risks associated with the study, we have now addressed this concern by including explicit details regarding the specific risks involved in the study as follows:
2.1. Study Design and Patients
“There was a potential risk of injury or falls due to excessive effort during the clinical assessments. To minimize the risk, the examiner provided detailed explanations regarding the clinical assessment procedures. Furthermore, vigilant monitoring of the patient was consistently maintained to promptly identify any indications of excessive effort or potential hazards associated with the assessments.” (page 3, line 106-110)
Comment 4
As you say in the limitation, the sample size is really small.
>> Response
We concur with the reviewer's observation regarding the limited sample size in our study. We have taken great care to provide a careful and conservative interpretation of the results throughout the Discussion and Limitations sections. In the Conclusions, we have incorporated the following sentence to address this concern:
- Conclusions
“Further study with a greater sample size is needed to clarify the relationship between preoperative motor function and postoperative HRQOL.” (page 11, line 273-275)
Comment 5
Results: In all the Tables add the IC and use bold to sign the significant results; add the adjustments that have been made.
>> Response
In response to the reviewer's suggestion, we have included 95% confidence intervals (CIs) and highlighted significant differences using bold signs in Tables 2-5.
Comment 6
Discussions: why is table 5 in this section? could be better in results. However, some studies are dated.
>> Response
As per the reviewer's recommendation, we relocated the table to the Method section, where we introduced the independent variables for the regression models.
Comment 7
References: the bibliography is dated and should be updated. It would be useful to include the references of these articles for international comparisons:
>> Response
We express our gratitude to the reviewer for providing valuable reference suggestions. We have incorporated these recommendations into the reference list, ensuring that the additional references, including those proposed by the reviewer, have been included and appropriately updated in the revised manuscript.

Reviewer 2 Report
Comments and Suggestions for Authors
Thank you for your manuscript. Your topic is interesting. However, you need quite work to improve your manuscript. Please see my comments below…
TITLE
I suggest changing the title according to the aim of the study: “Preoperative Motor Function is Associated with Short-term 2 Gain of Quality of Life After a Lumbar Surgery in Lumbar Degenerative Disease”.
INTRODUCTION
According to the aim of the study, you wanted to evaluate patients with lumbar degenerative disease. However, no information about this disease (e.g., description, epidemiology) is presented in introduction section.
According to the aim of the study, you wanted to evaluate the motor function of patients with lumbar degenerative disease. However, it is not clear during Introduction and Material and Methods sections how motor function can be evaluated neither why some tests were chosen for this purpose.
METHODS
It was not clear when were evaluated the various parameters, i.e., pre and post-surgery.
It was not clear during Material and Methods section that being a patient with lumbar degenerative disease was an inclusion criterion.
P2L88. “2.3. Predictors” – Are all the presented parameters predictors? Moreover, during introduction and Materials and Methods section the importance of some parameters/tests for the study it is not clear.
P2L90. “…weight…” – The term correct is body mass.
RESULTS
P2L90. “Patient recruitment flow is shown in Fig.1. We initially included 108 subjects admitted for lumber surgery during the study period. We excluded 56 patients based on our exclusion criteria.” – This sentence repeats what was explain in Materials and Methods section.
Author Response
Reviewer 2
Thank you for your manuscript. Your topic is interesting. However, you need quite work to improve your manuscript. Please see my comments below…
>> Response
We would like to express our gratitude to the reviewer for conducting a thorough review and for providing constructive comments and suggestions. In response, we have addressed the comments and highlighted the changes in red within the revised manuscript. Please find our detailed point-by-point response to each comment below.
Comment 1
TITLE: I suggest changing the title according to the aim of the study: “Preoperative Motor Function is Associated with Short-term 2 Gain of Quality of Life After a Lumbar Surgery in Lumbar Degenerative Disease”.
>>Response
As suggested by the reviewer, we changed the title as follows:
“Preoperative motor function is associated with short-term gain of quality of life after a surgery for lumbar degenerative disease: a pilot prospective cohort study in Japan”.
Comment 2
INTRODUCTION: According to the aim of the study, you wanted to evaluate patients with lumbar degenerative disease. However, no information about this disease (e.g., description, epidemiology) is presented in introduction section.
>> Response
We thank the reviewer for raising an important point. As suggested by the reviewer, we added general background of the lumber degenerative disease in the Introduction as follows (page 1, line 32-39):
“Lumbar degenerative disease, also known as Degenerative Lumbar Spine Disease, encompasses conditions such as Lumbar Disk Herniation (LDH), Lumbar Spinal Ste-nosis (LSS), and Degenerative Spondylolisthesis (DS). Globally, over 266 million indi-viduals have been diagnosed with lumbar degenerative disease [1] . This condition not only manifests as lower back pain but can also give rise to motor impairments [2], gait instability [3], and diminished physical activity [4]. Consequently, these symptoms contribute to a reduction in activities of daily living (ADL) and a decline in health-related quality of life (HRQOL).”
Comment 3
According to the aim of the study, you wanted to evaluate the motor function of patients with lumbar degenerative disease. However, it is not clear during Introduction and Material and Methods sections how motor function can be evaluated neither why some tests were chosen for this purpose. 
>> Response
We appreciate the reviewer for raising an important point. We added the following sentences to the Introduction and Materials & Method sections.
- Introduction
“However, the majority of prior investigations into preoperative and postoperative patient characteristics have predominantly relied on patient-reported out-comes (PROs). Standard PROs such as EQ-5D and ODI have been extensively used in various registries, such as the Swedish Spine Registry [13] and the Norwegian Registry for Spine Surgery [14]. Although these PROs enable clinicians to evaluate client’s perspec-tive, they are susceptible to response bias [15] and may not accurately reflect the client’s motor function [2]. To comprehensively capture the patient’s condition, it is important to complement PROs with clinical assessments based on objective evaluations by clinicians. Utilizing measures such as the Five Times Sit to Stand Test (FTSTS) and the Timed “Up and Go” test (TUG) can provide a more holistic understanding of the patient’s condition, addressing the limitations associated with PROs [16]. Lumbar degenerative disease results in diverse motor impairments, including paraspinal muscle atrophy [17], diminished static [18] and dynamic balance function [19, 20], and reduced walking endurance [21]. Previous studies have employed clinical evaluations such as the TUG [19], FTSTS [20], and 6-minute walk distance (6MWD) [21] to objectively assess motor impairments in in-dividuals with lumbar degenerative disease. These assessment tools are not mutually exclusive, as they collectively reflect the client’s mobility. However, each assessment may elucidate different aspects of subsystems contributing to overall mobility. Despite these prior investigations, the relationship between preoperative assessment outcomes and postoperative recovery of HRQOL in individuals with lumbar degenerative disease remains unclear [19, 20, 21].” (page 2, line 65-85)
2.3. Exposure
“Clinical data included intensity of back and leg pain; duration of symptoms [29]; Oswestry Disability Index (ODI) [30]; Self-rating Depression Scale (SDS) [31, 32]; 10 Meter Walking Test; Timed “Up and Go” Test (TUG) at comfortable gait speed [33, 19]; 6MWD [21]; Knee Extension Strength with Handheld Dynamometer; FTSTS [20]. Preoperative ODI score is used as a predictor of EQ-5D score postoperative 12 months in patients with lumbar degenerative disease [34]. Similarly, preoperative SDS scores predict achievement of MCID for SF-36 at 12 months postoperatively [32]. We used TUG [19] and FTSTS [20] as the ob-jective measures of motor function. These assessments provide insight into objective functional impairments by encompassing motor tasks commonly challenging for individ-uals with lumbar degenerative disease, such as gait and sit-to-stand transitions [19, 20].” (page 3-4, line 134-143)
Comment 4
METHODS: It was not clear when were evaluated the various parameters, i.e., pre and post-surgery.
>> Response
For better clarity, we revised the description as follows:
“Demographic, medication, surgical, and clinical data were collected at 1-3 days before the surgery” (page 4, line 126-127)
Comment 5
It was not clear during Material and Methods section that being a patient with lumbar degenerative disease was an inclusion criterion.
>> Response
We added the following line to clarify the inclusion criteria.
“Patient selection flow is shown in Fig. 1. In this prospective cohort study, we initially included 108 patients who underwent surgery for lumbar degenerative disease at a gen-eral hospital in Japan between May 2021 and October 2021.” (page 3, line 95-97)
Comment 6
P2L88. “2.3. Predictors” – Are all the presented parameters predictors? Moreover, during introduction and Materials and Methods section the importance of some parameters/tests for the study it is not clear.
>> Response
We did not use all the parameters as predictors in our regression models. For better clarity, we changed the section heading from Predictors to Exposures. Furthermore, in the section 2.4. Statical Analysis, we clarified how and why we chose specific independent variables in our regression models. We also clarified the importance of the parameters used in our study in the section 2.3. Exposure.
2.3. Exposures
“Clinical data included intensity of back and leg pain; duration of symptoms [29]; Oswestry Disability Index (ODI) [30][16 Roland 2000]; Self-rating Depression Scale (SDS) [31, 32]; 10 Meter Walking Test; Timed “Up and Go” Test (TUG) at comfortable gait speed [33, 19]; 6MWD [21]; Knee Extension Strength with Handheld Dynamometer; FTSTS [20]. Preoperative ODI score is used as a predictor of EQ-5D score postoperative 12 months in patients with lumbar degenerative disease [34]. Similarly, preoperative SDS scores predict achievement of MCID for SF-36 at 12 months postoperatively [32]. We used TUG [19] and FTSTS [20] as the objective measures of motor function. These assessments provide insight into objective functional impair-ments by encompassing motor tasks commonly challenging for individuals with lum-bar degenerative disease, such as sit-to-stand transitions and gait [20, 35].” (page 3-4, line 134-143)
2.4. Statical Analysis
“The independent variables of the Model 1 were selected according to the clinical relevance judged by researchers based on the results of previous studies (Table 1). We added FTSTS to the existing models to infer the impact of pre-operative motor function to the outcome.” (page 4, line 156-159)
Comment 7
P2L90. “…weight…” – The term correct is body mass.
>> Response
We revised the sentence accordingly. (page 3, line 127)
Comment 8
RESULTS: P2L90. “Patient recruitment flow is shown in Fig.1. We initially included 108 subjects admitted for lumber surgery during the study period. We excluded 56 patients based on our exclusion criteria.” – This sentence repeats what was explain in Materials and Methods section.
>> Response
We thank reviewer for pointing out the redundant sentence. We removed the duplicate from the Result section in the revised manuscript.

Round 2
Reviewer 1 Report
Comments and Suggestions for Authors
Congratulations for your work. I think it is also useful to add the last recommended article (Fineschi D, Acciai S, Napolitani M, Scarafuggi G, Messina G, Guarducci G, Nante N. Game of Mirrors: Health Profiles in Patient and Physician Perceptions. International Journal of Environmental Research and Public Health. 2022; 19(3):1201. https://doi.org/10.3390/ijerph19031201) as it talks about the empathic doctor-patient relationship.
Reviewer 2 Report
Comments and Suggestions for Authors
No comments.